# Hydrocarbon Hydrate Flow Assurance History as a Guide to a Conceptual Model

**DOI:** 10.3390/molecules26154476

**Published:** 2021-07-24

**Authors:** E. Dendy Sloan

**Affiliations:** Department of Chemical and Biological Engineering, Colorado School of Mines, 1500 Illinois St, Golden, CO 80401, USA; esloan@mines.edu

**Keywords:** history, flow assurance, best practices, restart, blockage, model

## Abstract

This work reviews major hydrocarbon hydrate advances in flowline applications of 25 international hydrate organizations. After a review of hydrate history and the current state-of-the-art, four conclusions were drawn: (1) engineers must take risks and cannot always afford the luxury to await scientific developments, (2) industry is more likely than academia to suggest hydrate needs and solutions, (3) the best hydrate blockage prevention practices are evolving and (4) a stepwise conceptual model can be proposed for a transient restart flowline hydrate blockage.

## 1. A Brief Hydrate History

### 1.1. Curiosity

Curiosity and intellectual interest were the initial clathrate hydrate motivators. Hydrate discovery is usually credited to Sir Humphrey Davy [1], the mentor of Michael Faraday, in 1811. Earlier, hydrate discoverers, such as Joseph Priestley [2] did not provide reproducibility of their experiments above 273 K, to ensure the solid discovered was not ice. European researchers discovered clathrate hydrates of natural gas and oil mixtures, as summarized in Table 1. Notably, the perseverant laboratories of de Forcrand and of Villard (1882–1925) discovered several clathrate hydrates of small hydrocarbons, such as some components of natural gas: CH_4_, C_2_H_6_, C_2_H_4_, C_3_H_8_, N_2_ and H_2_S.

After 1925, X-ray diffraction was used to determine hydrate structures. After two decades of X-ray data, interpretation by von Stackelberg and co-workers [3,4,5,6], Claussen [7,8], and Pauling and Marsh [9] defined two hydrate crystal structures (sI and sII). Both structures are composed of the largest regular polyhedron, the pentagonal dodecahedron (5^12^) of water molecules, i.e., the basic “hydrate building block” cage containing 12 pentagon faces of hydrogen-bonded water molecules. The 5^12^ cavity is attached to other 5^12^ cavities through the 5^12^ vertices to compose sI (with 5^12^6^2^ cages, having two hexagonal faces in addition to the 12 pentagonal faces), or through the 5^12^ faces to compose sII, (with 5^12^6^4^ cages). A unique hydrate feature is that the guest molecule is trapped inside each water cage via mostly repulsive van der Waals forces, without chemical or hydrogen bonds. A pure component guest molecule smaller than n-pentane is required to prop open the water cavities and form a hydrate structure; however, mixtures can form with larger molecules. In 1952, all existing hydrocarbon hydrate structures were thought to be sI or sII. Over three decades later, a new hydrate structure (sH) was discovered by Ripmeester, et al. [10], having been overlooked in the previous data. Details of the hydrate crystal structures are in chapter two of Sloan and Koh [11].

Today scientific curiosity continues to be a major driving force in hydrate research. Rather than listing them here, scientific curiosity advances are interspersed with the following application timelines, as the hydrate history unfolds. Although there were only 22 clathrate hydrate publications from 1811 to 1834, the time of their discovery in hydrocarbon flow channels, intellectual interest and curiosity remain today as principal motivating factors in hydrate research.

**Table 1 molecules-26-04476-t001:** Hydrates from 1810 to 1934. Abstracted from pg. 4 of Sloan and Koh [11].

Year	Event
1811	Chlorine hydrate discovery by Sir Humphrey Davy
1823	Corroboration by Faraday-proposed formula Cl_2_•10H_2_O
1882	De Forcrand suggested hydration number H_2_S•(12-16)H_2_O and measured 30 binary hydrates of H_2_S with a second component such as CHCl_3_, CH_3_Cl, C_2_H_5_Cl, C_2_H_5_Br, C_2_H_3_Cl. He indicated all compositions as G•2H_2_S•23H_2_O where G = 2nd guest molecule (other than H_2_S)
1884	Le Chatelier showed the Cl hydrate P–T curve changes slope at 273 K
1884,5	Roozeboom postulated lower/upper hydrate quadruple points (Q_1_ = I-L_w_-H-V, Q_2_ = L_w_-H-V-L_HC_), using SO_2_ as evidence; determined univariant dependence of P on T
1888	Villard obtained the temperature dependence of H_2_S hydrates
1888	De Forcrand and Villard measured temperature dependence of CH_3_Cl hydrate
1888	Villard measured hydrates of CH_4_, C_2_H_6_, C_3_H_8_, C_2_H_2_, N_2_O
1890	Villard measured hydrates of C_3_H_8_ and suggested that the temperature of the lower quadruple. Point (Q_1_) decreased by increasing the molecular mass of a guest; Villard suggested hydrates were regular crystals
1896	Villard measured hydrates of Ar and proposed that N_2_ and O_2_ form hydrates; Used heat of formation data to get the water/gas ratio
1897	De Forcrand and Thomas sought double (w/H_2_S or H_2_Se) hydrates; found mixed (other than H_2_S_x_) hydrates of numerous halohydrocarbons mixed with C_2_H_2_, CO_2_, C_2_H_6_
1902	De Forcrand first used Clausius–Clapeyron relation for ΔH and compositions; tabulated 15 hydrate conditions
1919	Scheffer and Meyer refined Clausius–Clapeyron technique as applied to hydrates

### 1.2. Flow Assurance: From Apprehension to Avoidance to Management

Before 1934, it was thought that hydrocarbon–water flowlines were being blocked by an unusual crystal substance, attributed to various causes with a great deal of apprehension, but without much definition. Hammerschmidt [12,13] studied the German monograph of Schroeder [14] to initially determine that the water + gas, crystalline flowline plugs above the ice point were clathrate hydrates. Hammerschmidt also created a simple equation to predict the pressure–temperature (P–T) formation conditions and determined many thermodynamic hydrate inhibitors, including those commonly used today, methanol and mono ethylene glycol (MEG). The Hammerschmidt equation, still reliable as an initial first inhibition estimate, allowed the natural gas industry to change the paradigm, from one of apprehension to the avoidance afforded by thermodynamic pressure, temperature and inhibitor concentration predictions.

In 1941, Katz determined that, unlike many other natural gas pure solid precipitates, clathrate hydrate conditions could be predicted as ideal solid solutions of natural gas components. Over the next decade, this discovery motivated the Katz laboratory to generate a series of vapor–solid equilibrium charts for natural gas components, commonly called K_vsi_ charts, where individual component K_vsi_ (≡y_i_/z_i,_ the water-free ratio of component i mole fraction in the vapor y_i_ and solid z_i_) values were a function of temperature and pressure. The use of these individual component K_vsi_ charts is like the more common vapor-liquid K_i_ charts, where K_i_ (≡y_i_/x_i_, the ratio of component i mole fraction in the vapor y_i_ and liquid hydrocarbon x_i_). These K_vsi_ charts enabled the hand calculation of the temperature and pressure of the solid hydrate plug formation and the solid dewpoint (where Σ y_i_/K_i_ = 1). Examples of the use of both the Katz K_vsi_ charts and the Hammerschmidt equation are provided in pp 215–233 of Sloan and Koh [11].

It was a sincere pleasure to dine with Professor Katz in 1984, who discussed what was to evolve into the first conclusion of this work. When Professor Katz was asked, “Because many of the other natural gas precipitates were known as pure solids (e.g., ice, CO_2_ and H_2_S) how did you arrive at the idea in 1940 that hydrates were ideal solid solutions? You didn’t have the crystal structures, solid compositions, spectroscopic information, or a statistical thermodynamic model”, Professor Katz replied, after a moment, “You know, when one doesn’t have such sophisticated tools, one just has to think!”.

The above, startling response was not only a modest indication to Professor Katz’s outstanding intuition, but also his denigration of the misbelief that engineering is only applied science. As Koen [15] suggests in his definition of the engineering method, often the science is not available to be applied, and engineers must take considered risks before the science become available, with chances of success or failure. Engineering risk-taking is frequent, particularly in the Information Technology (IT) industry. Very many successful IT entrepreneurs, have previously failed and learned from their failures. 

It was only when hydrates were discovered inside [12] and outside [16] flowlines that industrial interest expanded beyond academic curiosity. Because there are many more industrial practitioners than academics, practical applications drive both interest and publications. Industry is more likely than academia to suggest hydrate needs and solutions. This second conclusion is illustrated by the exponential growth in the number of publications per decade in Figure 1; in each decade, the number of hydrate publications increased by an average factor of 2.5 in the twentieth century. The semi-logarithmic plot of Figure 1 is not quite linear; the slope increases in 1934, the year of hydrate discovery in flowlines, and again in 1965, the year of hydrate discovery in nature. 

Figure 1 provides evidence that industry is more likely than academia to suggest hydrate needs and solutions, because energy applications motivate industrial interest. Both hydrate applications increased the publication rate due to the need for energy, one of the largest drivers of a national gross domestic product (GDP), which is one measure of national economic success. Over the 30-year life of the triennial International Conference on Gas Hydrates, interest has shifted to the point that more than 80 percent of attendees are interested primarily in hydrated energy recovery.

### 1.3. A Statistical Theory of Hydrate Thermodynamics

It was only after the determination of hydrate structures I and II, composed of singly occupied guests in well-defined water cavities, that a statistical theory was generated for the solid hydrate phase, by van der Waals and Platteeuw [17]. The model was first successfully applied by the Kobyashi Lab [18] to gas mixtures, and then to natural gas mixtures by Parrish and Prausnitz [19]. The model was almost too good, because, for a number of years thereafter, the hydrate phase composition was predicted by the model, rather than measured.

After X-ray crystallography enabled the definition of the hydrate crystal structures, measurements of the hydrate phase occupancy awaited spectroscopic measurements, beginning with NMR by Davidson and colleagues [20] and Raman measurements by Sum et al. [21]. These measurements showed three small errors in the van der Waals and Platteeuw model: (1) guest molecule stretch cages; (2) water molecules beyond the first shell in each cage contribute to the chemical potential; and (3) cage radii vary with temperature, pressure and equilibrium fluid compositions. Such corrections allow for a thermodynamic prediction of hydrate formation pressures and temperature to within 10% and 1K, respectively.

### 1.4. Beyond Thermodynamics to Kinetics: From Avoidance to Management

In 1980, Bishnoi and colleagues began a series of kinetic studies of hydrate formation [22]. Like other time-dependent studies, at least an order of magnitude of accuracy was sacrificed, relative to thermodynamic, time-independent studies. One particularly important result was recently shown by Ripmeester [23] to summarize some of the kinetic data in three laboratories (Canadian NRC, U. Göttingen, and GFZ Potsdam). Data in Figure 2 show that solid-phase kinetics are extremely slow; small amounts of meta-stable hydrate phases persist for long duration. Nevertheless, kinetics results permitted the hydrate paradigm change, from (1) Apprehension to (2) Avoidance and, finally, to (3) Management (Sloan [24]).

### 1.5. Modern Hydrate Advances

Table 2 summarizes some of the modern hydrate flow assurance developments. Due to space limitations, the advances in Table 2 are listed, not discussed; a thorough discussion of each development would likely require individual monographs. 

### 1.6. The Evolution of Best Practices for Hydrate Flow Assurance

Equinor (formerly Statoil) has explicitly complied best engineering practices for hydrate flow assurance, initially by Kinnari et al. [25]. The concept of “best practices” is very important in engineering, not only because they represent years of engineering experience, but also because best practices determine litigation outcomes. Hydrate flow assurance best practices fall into five broad categories shown in Figure 3:Process Solutions: (a) remove the water and (b) dehydrate the gas.Hydraulic Methods: (a) dense phases, (b) compression, (c) depressurization, (d) gas sweep and (e) fluid displacement.Thermal Methods: (a) Insulation, (b) direct electrical heating, (c) pipe bundles and d) heat tracing.Chemical Methods: (a) alcohols, (b) glycols, (c) low dosage inhibitors (KHIs and AAs) and (d) salt.No Hydrate Control Measures: (a) low amounts of subcooling, (b) natural kinetic growth inhibition and (c) natural transportability methods.

The rightmost portion of Figure 3 represents the evolution of hydrate control methods. as kindly provided by Equinor (Li et al. 2019). A comparison of both the left and right portions of Figure 3 suggests two things: (a) the above five basic categories of flow assurance are still appropriate; and (b) industry is trending toward safe operation in the hydrate domain, using modern tools which will allow management of hydrate formation to prevent blockages. In addition to the five major control methods above, new modeling methods go beyond thermodynamic equilibrium, using time-dependent kinetic phenomena. As a result, the paradigm for hydrate flow assurance has had two major shifts: (1) first from apprehension to thermodynamic avoidance in 1934, and (2) a second time from thermodynamic avoidance toward kinetic management, beginning around 2000.

## 2. Conceptual Stages of Hydrate Plug Formation on Transient Restart

With the evolution of experiments in the laboratory, the pilot flowloops and with field experiments, conceptual models have arisen which might enable hydrate flow assurance. As one example, the following conceptual word picture is an effort to synthesize transient hydrate laboratory, flowloop and field experiments over a number of decades. The experimental data for much of the following conceptual picture are summarized in theses from the Colorado School of Mines, particularly the theses of Pickarts [27] and Ismail [28].

Like most conceptual syntheses, some of the details are perhaps incorrect. For those potential errors and unintended slights of other laboratories, the author apologizes in advance for choosing the most familiar experimental evidence. However, the evidence seems sufficient to synthesize an initial conceptual picture for transient startup hydrate blockage formation in a low surfactant oil and gas flowline.

In normal offshore flowline operation, hydrates do not form, due to temperatures, pressures and concentrations outside hydrate thermodynamic conditions. Reservoir fluids, including progressive water amounts, are at sufficiently high temperature and pressures, so production will reach the platform as fluids, frequently aided by flowline insulation or inhibitor injection at the wellhead. Nevertheless, substantial heat is transferred from the flowline to the surrounding water at ~277 K, typically below 1200 m of water depth.

When flow stops, for example, due to failure of platform equipment, e.g., a compressor, a separator or a dehydrator, there is a “no touch” time of about half a day, while the platform repair process attempts to resume steady state operation. During this time, the pressurized flowline cools, approaching the hydrate stability region at the seafloor temperature of 277 K.

For flow interruptions longer than the “no touch” time, efforts will be made to prevent flowline hydrate formation, for example, using a fluid displacing dead oil in the pipeline (“bull heading”), or by depressurizing the pipeline to remove it from the hydrate pressure at 277 K. The following scenario for hydrate formation suggests a conceptual picture of what will happen if hydrates form before the flowline restarts.

When flow stops, the phases separate and pool as gas, oil and water according to density. The low-density gas (<320 kg/m^3^) is at the flowline top, oil is in the middle (mineral oil 70T has a typical density of 780 kg/m^3^) and water with a pure water density of 1000 kg/m^3^ is at the bottom. Local flowline low spots encourage phase pooling, which may not represent the overall flowline conditions.

As the pipeline subcools about 3.5 K into the hydrate stability condition, hydrates (density ~ 900 kg/m^3^) form an initial thin (~10 µm thick) film at the oil–water interface. Because flowline hydrates are 85 mol% H_2_O with molecules smaller than C_5_H_12_, three things happen: (1) water-in-oil emulsions frequently break (Høiland et al. [29]); (2) hydrate formation consumes molecules smaller than C_5_H_12_ dissolved in the oil phase, requiring subsequent dissolution and diffusion of small gas molecules into the oil layer to reach the hydrate film at the oil–water interface; and (3) a hydrate thin film covers the water phase at the hydrocarbon interface, so that the pseudo-solid is initially as little as 4 volume per cent hydrate (Austvik [30], but anneals to a more solid mass.

Annealing initially occurs by water (not hydrocarbon) perfusion through the cracks in the hydrate film (Davies et al., [31]. The thin hydrate film is more solid than fluid, but initially in a transitional, malleable state which solidifies with time. When the flowline restarts, if sufficient time has passed for a solid hydrated mass to block the channel, the flow will stop.

However, if the hydrate is still a thin film, high startup turbulence will shear the film to form small hydrate-film-encrusted water droplets, which quickly cohere to form a larger and porous solid. Unconverted water remains as a separated layer, partitioned by the hydrate mass from the oil. Any free-water layer helps maintain movement of agglomerated hydrates. Over time, the unconverted water is infused into the hydrate mass until the free-water phase disappears. 

With the disappearance of the free-water layer, the hydrate mass contacts the pipe wall as a deposit. Even with a small amount of hydrate and high porosity (85–90%), the deposit may impede flow. With flow stopped, the mass will further anneal to a more substantial hydrate solid. At that point, corrections must be taken, such as depressurization, inhibitor injection with coiled tubing, etc.

## 3. Conclusions

Because flow assurance academic researchers are greatly outnumbered, an industrial majority of flow assurance professionals have determined many of the hydrate flow assurance solutions which have grown into hydrate prevention best practices, for example, as illustrated in Figure 3.

Over two centuries since the discovery of hydrates, experimental evidence has evolved—for example, enabling one new conceptual picture of transient restart hydrate formation in oil and gas pipelines presented at the conclusion of this work. Like most syntheses, some details will either be disputed or considered inadequate. However, the transient hydrate formation restart conceptual picture draws together much of the current experimental evidence from laboratories, flowloops and field data.

Some of the details in the transient restart hydrate formation concept require scientific verification. However, the transient restart concept may enable better flow assurance. As a minimum, the concept might serve as a basis for future corrections, considering Francis Bacon’s dictum, “Truth emerges more readily from error than from confusion”.

## Figures and Tables

**Figure 1 molecules-26-04476-f001:**
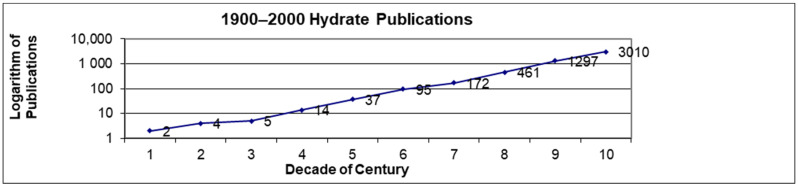
Base 10, semi-logarithmic plot of the number of hydrate publications by decade, from 1900–2000. Note the increases in slopes about 1934 and 1965, the years of hydrate discovery inside and outside of flowlines, respectively.

**Figure 2 molecules-26-04476-f002:**
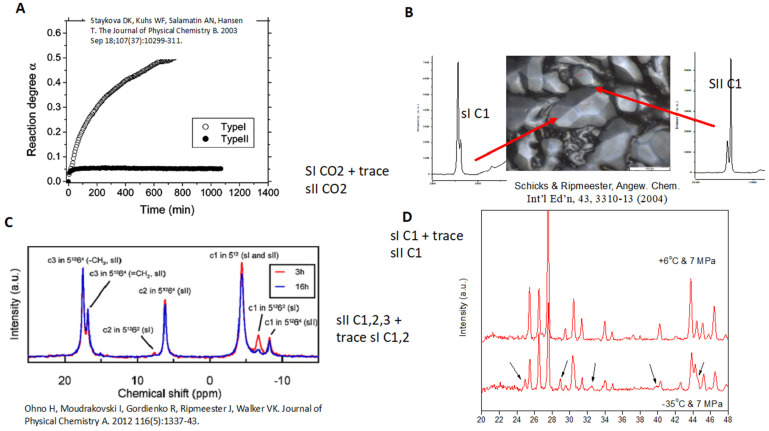
Spectroscopic data from Canadian NRC (**C**,**D**), U. Göttingen (**A**) and GFZ Potsdam (**B**), showing metastable hydrate phases after two days (Ripmeester [23]).

**Figure 3 molecules-26-04476-f003:**
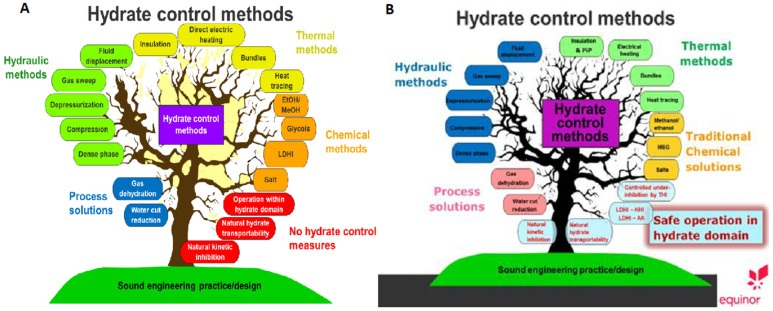
Equinor’s concept of hydrate flow assurance best practices in 2014 (**A**) and 2019 (**B**). Note the lower rightmost methods, are trending toward a “do nothing” strategy (Li, et al., [26]).

**Table 2 molecules-26-04476-t002:** Modern hydrate flow assurance developments.

Year	Events
1966	NMR measurements of the hydrate phase by Davidson and Ripmeester
1980	Kinetics study begun (Bishnoi et al.)
1982	1st flowloop constructed (Sintef in Norway)
1987	New structure H (sH) hydrates discovered (Ripmeester et al.)
1995	Kinetic inhibitors (KHI) used in North Sea (BP)
1996	Raman measurements of hydrates (Colorado School of Mines[CSM])
1999	Depressurization plug removal model (CSM)
1990’s	Extended tiebacks eliminated tension leg platform need (DeepStar)
1999	Hydrates declared major deep water problem for flow assurance (DeepStar)
2000	Anti-agglomerates used in Gulf of Mexico for Water Cuts (WC) < 50% (Shell)
2001	Initial kinetics model enable change from avoidance to management (Shell)
2002	For P < 275 bara prediction accuracy is to within 1K and 10% P
2003	Formation of plug incorporated in flow simulators (OLGA)
2003	Very slow (>1000 min) conversion of metastable hydrate structure (Göttingen, Potsdam, NRC)
2003	Cold Flow (BP, XoM)
2007	Acoustic plug locator (Heriot–Watt)
2008	N_2_ used for plug removal (BP)
2009	Formation risk monitoring and detection methods (Heriot–Watt)
2010	Electrical heating for plug removal
2010	Hydrate plug resistant oil protocols (Petrobras, Shell)
2012	KHI recovery and reuse methods (Heriot–Watt)
2012	Hydrate flowline deposition is important addition to aggregation (XoM)
2014	Best practices established for prevention/removal (Statoil/Equinor)

## Data Availability

No new data were created or analyzed in this study. Data sharing is not applicable to this article.

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
