# Peer review of "Hydrocarbon Hydrate Flow Assurance History as a Guide to a Conceptual Model"

_molecules, 2021, doi:10.3390/molecules26154476_

Round 1

Reviewer 1 Report

This paper is a brief, but very interesting review documenting the history of hydrate advances starting from scientific curiosity to engineering practices in hydrate flow assurance. It would be good if the author can add a figure showing the transient hydrate formation restart model to help readers understand, but the paper itself is good enough as it is. 

Author Response

The author thanks this reviewer for her/his considerate and thoughtful review.  The reviewer is correct; the transient restart conceptual model which concludes the manuscript is the result of a career of research, done mostly by graduate and postdoctoral students.  The suggestion of a figure supporting the model is interesting; however, the conceptual model would require over six figures to express it in detail, and so the figures have not been included.  Over the next decade of research, some of these conceptual details will be refined in other laboratories and in ours.

Reviewer 2 Report

I really enjoyed and learned a lot reading this remarkable piece of work where more than 200 years of hydrate flow assurance history is condensed effectively in such a small space.

The transient start concept given at the end is cherry on top. It will definitely catch the attention of academic researchers and will open a field of study where finding the missing ends or verification of the proposed information will be at the forefront.

I recommend the publication of this mini review in its current form.

Author Response

The author thanks this reviewer for her/his considerate and thoughtful review.  The reviewer is correct; the transient restart conceptual model which concluded the manuscript is the result of a career of research, done mostly by graduate and postdoctoral students culminating in this conceptual model.  The reviewer's conclusion that the details of the conceptual picture require future verification is true and that was the intention of the manuscript, which I understand is to be published first in this special hydrate issue of the journal.